# Latitude and Weather Influences on Sun Light Quality and the Relationship to Tree Growth

**Camilo Chiang [1,2,*]** , **Jorunn E. Olsen [3]**, **David Basler [4]**, **Daniel Bånkestad [2]** and **Günter Hoch [1]**

1   Department of Enviromental Siences – Botany, Schönbeinstrasse 6, 4056 Basel, Switzerland
2   Department of Research and Development, Heliospectra, Box 5401,414 58 Göteborg, Sweden
3   Department of Plant Sciences, Faculty of Biosciences, Norwegian University of Life Sciences, P.O. Box 5003, N-1432 Ås, Norway
4   Department of Organismic and Evolutionary Biology, Harvard University, Cambridge, MA 02138, USA
*   Correspondence: Camilo.chiang@unibas.ch; Tel.: +46-727-764-880

**Abstract:** Natural changes in photoperiod, light quantity, and quality play a key role in plant signaling, enabling daily and seasonal adjustment of growth and development. Growing concern about the global climate crisis together with scattered reports about the interactive effects of temperature and light parameters on plants necessitates more detailed information about these effects. Furthermore, the actual light emitting diode (LED) lighting technology allows mimicking of light climate scenarios more similar to natural conditions, but to fully exploit this in plant cultivation, easy-to-apply knowledge about the natural variation in light quantity and spectral distribution is required. Here, we aimed to provide detailed information about short and long-term variation in the natural light climate, by recording the light quantity and quality at an open site in Switzerland every minute for a whole year, and to analyze its relationship to a set of previous tree seedling growth experiments. Changes in the spectral composition as a function of solar elevation angle and weather conditions were analyzed. At a solar elevation angle lower than 20°, the weather conditions have a significant effect on the proportions of blue (B) and red (R) light, whereas the proportion of green (G) light is almost constant. At a low solar elevation, the red to far red (R:FR) ratio fluctuates between 0.8 in cloudy conditions and 1.3 on sunny days. As the duration of periods with low solar angles increases with increasing latitude, an analysis of previous experiments on tree seedlings shows that the effect of the R:FR ratio correlates with the responses of plants from different latitudes to light quality. We suggest an evolutionary adaptation where growth in seedlings of selected tree species from high latitudes is more dependent on detection of light quantity of specific light qualities than in such seedlings originating from lower latitudes.

**Keywords:** light quantity; light quality; spectrometer; shoot elongation; tree seedlings

---

## 1. Introduction

Light is one of the main environmental signals affecting plant biology, with multiple physiological responses being controlled by changes in light quantity, quality, and photoperiod [1–3]. Although the effects of the natural daily variation in light quality on plants have not been quantified in situ, it could be shown experimentally by using artificial lighting of defined wavelength ranges, that specific developmental processes in plants are differently affected by different fractions of the sunlight spectrum [4–7]. Punctual measurements comparing sun light spectral composition at different solar elevation angles showed a lower fraction of blue (B, 400–500 nm) and red (R, 600–700 nm) light and a higher fraction of green (G, 500–600 nm) light in the middle of the day than at sunset [1]. Smith et al. [1] quantified the effect of the weather conditions on light quality at high solar elevation angles

and showed that clouds and dust cover have a small effect on the light spectra, mainly affecting the light in the B and R ranges, not unlike the changes in the spectral composition of sun light that occur when passing through a plant canopy. Yet, detailed information about the dynamic changes in these light qualities, especially with respect to their potential impact on plant biology, have so far not been reported, although there is substantial knowledge about static light quality effects on gas-exchange and other plant physiological processes [8–10].

At the short wavelength end of the sun spectra, effects of ultraviolet (UV) light on plants (e.g., shoot elongation, production of UV-protecting secondary compounds) and the associated UV-B receptors (UVR8) and UV-A-blue light receptors, have been well described in plants [4]. Interestingly, the signaling effects of UV light on plants is reduced at higher radiation, implying that UV as a plant signal may be most important during twilight. The next section of the light spectrum, the B light, which is mainly sensed by cryptochromes, phototropins, and other blue light-UV-A receptors, affects, for example, stomatal opening and plant phototropism. High percentages of B light have been shown to affect plant morphology [5]. Although chlorophyll, as the central plant pigment of the photosynthetic light reaction, absorbs mainly B and R light, G light has also been shown to contribute to photosynthesis and to be especially important at lower canopy levels (i.e., the so-called 'green shade') and at deeper levels of the leaves [6]. At the longer wavelength end of the visible sun spectra, R and far-red (FR) light have important signal functions for plants. The ratio between red to far red (R:FR) is sensed by the phytochrome system and changes in the R:FR ratio can influence important physiological processes like growth, germination, and flowering. In addition, FR has an important role in optimizing photosynthesis upon combined action of the PSII and PSI, increasing the photosynthetic efficacy [7].

Recent developments in light emitting diode (LED) lighting systems potentially enable the mimicking of more natural light quality changes during plant cultivation in indoor growth facilities [11]. Due to the high degree of absorption of B and R light by photosynthesis-related pigments and higher electric efficiency [8], these two wavelength ranges tend to be dominating in commercial LED lamp systems. However, the knowledge about the changes in light quality related to the solar elevation angle, latitude, time of the day, and the day of year, as well as the weather in general [1,12] has been so far reported mainly from an atmosphere-physical approach, and has not been transferred to actual lighting systems used for plant culture in greenhouses or growth chambers.

Changes of light quality in the morning and evening hours may be an especially important plant signal at higher latitudes where twilight conditions persist for a substantial period. Several studies have shown how different ecotypes of tree species react differently to R or FR light treatments as day extension, and it has been hypothesized that this could be due to adaptations to the light quality at the end of the day at their site of origin [13–15]. Additionally, it has been shown that light quality can interact with other environmental factors, like temperature, where higher temperatures have shown to reduce the promoting effect of FR light on growth [16]. Understanding the role of the light quality variation in plants is a crucial factor to predict the effect of the currently rising temperatures, especially in marginal areas such as those close to the latitudinal range limits of trees.

In the current study, we present detailed, easy-to-apply, and continuous field measurements of the natural changes of the spectral composition of sunlight over a full year at a mid-latitudinal site (47° N). This data was then combined with an analysis of studies investigating the effect of light quality on growth of tree seedlings of different latitudinal origin. Our study thereby focuses on the comprehensive effects of sun light quality changes due to weather conditions and time of the year, excluding further, smaller-scale modifications of the light spectra due to the presence of 'green shade' below a canopy. Here, we investigated the correlation between wavelength-specific light quantity requirements of tree seedlings from different latitude origins and the natural availability of these wavelengths due to geographical, annual, and diurnal changes, at their respective origin. Such a correlation would indicate ecotypic adaptations of tree populations to the specific spectral light quality and dynamics at their original site.

## 2. Materials and Methods

### 2.1. Light Spectra Recordings

A USB2000+XR1-ES spectrometer (25 μm entrance slit, range 200–1000 nm, 1.5 nm resolution, Ocean Optics Inc., Largo, Florida, USA) was installed twelve meters above ground level at the Botanical garden of the University of Basel (257 m AMSL, 47° 33′ 30.3″ N, 7° 34′ 52.4″ E, Basel, Switzerland) to acquire the light spectrum during a chronological year under a ensured shadow-free environment with minimalized light reflection from buildings, surface water bodies, or vegetation in the surrounding areas. Light spectra from 200–1000 nm were recorded every minute from 21 February 2018 to 21 February 2019 using a single board computer (Raspberry pi 2, Cambridge, UK) allowing dynamic change in the integration time, reducing the electric noise in the measurements through the use of 75–85% of the saturation point of the equipment. The optical fiber was installed at a 90° angle relative to the horizon. A cosine corrector made of Spectralon was used to capture environmental light coming from 180° (CC-3-UV-S, Ocean Optics Inc.). The cosine corrector was replaced every three months. The spectrometer was additionally equipped with a fan to avoid heat accumulation on hot days and was calibrated with a calibration lamp (HL-3 plus, Ocean Optics Inc.), once before mounting, and then every 3 months during the measurement year. The calibration lamp was warmed up for 15 min before the calibrations that were performed using a boxcar width of 2 wavelengths every 6 nm in both directions and the average of 5 measurements for each curve.

### 2.2. Light Energy Calculations

To acquire an initial dark library, the spectrometer was set in darkness at 20 °C, and a dark spectrum was recorded for integration times between 100 ms and 10 s every 100 ms. For each light measurement, the corresponding or interpolated dark spectrum was removed from the raw measurement in the corresponding integration time. The remaining count was multiplied with the corresponding calibration file of the calibration lamp. The resultant count was then divided by the area ($m^2$) of the cosine corrector ($1.19 \times 10^{-5}$ $m^2$) and then divided by the integration time of each measurement (s). Additionally, the energy in each particular wavelength was calculated multiplying each specific frequency by the Planck constant. To obtain the photon flux of each wavelength as μmol photons $m^{-2}$ $s^{-1}$, the resultant was divided by the Avogadro number and the pre-calculated energy of each wavelength [17].

### 2.3. Light Quantity and Quality Proportions

To simplify the results from a biological and practical point of view, from each measurement three proportions were separately calculated from the visible light spectra: The percentage of blue (B), green (G), and red (R). For the calculation of B, G, and R, the light spectrum as μmol photons $m^{-2}$ $s^{-1}$ was integrated from 400 to 700 nm to obtain the total photosynthetic photon flux density (PPFD). Furthermore, every 100 nm between 400 and 700 nm, the proportions of B, G, and R where calculated. The B proportion corresponded to the percentage of photons from 400 to 500 nm compared with the total PPFD, G from 500 to 600 nm and R from 600 to 700 nm, respectively. Additionally, the red to far red (R:FR) ratio was also calculated. This was done through the division of the sum of photons (as μmol $m^{-2}$ $s^{-1}$) between 655 and 665 nm and the sum of photons between 725 and 735 nm, respectively [18]. For the analysis of the weather conditions on the light spectra through the day, the sunniest and the most cloud-covered day of each month were selected from the recorded data (*n* = 12). After this, a locally estimated scatterplot smoothing (LOESS) regression was fitted for both weather conditions (i.e., clear sky and overcast).

### 2.4. Solar Elevation Angle Calculation

To quantify the average effect of the weather and remove the effect of the time of the day and day length through the year, the collected sun spectral data were analyzed as a function of the solar

elevation angle. For each measurement throughout the observation year, the solar elevation angle was calculated based on the geographic position and time of the day and the day of year using the solar position calculator available online [19] and confirmed through the OCE R package based on the NASA-provided Fortran program, using equations from "The Astronomical Almanac".

## 2.5. Literature Review

To relate our light quality measurements to potential effects on growth of tree seedlings from the boreal/temperate zone, we conducted a literature search and performed an analysis on a set of published experiments that investigated the effect of light quality on seedling growth of selected tree species from different latitudes: The conifers *Pinus sylvestris* L., *Picea abies* (L.) H.Karst, and *Abies lasiocarpa* (Hook.) Nuttall, as well as the deciduous *Betula pendula* Roth. (more information in Supplementary Table S1). We exclusively choose studies that (1) were conducted under similar controlled conditions, (2) treated tree seedlings with different R:FR ratio light, (3) made quantitative growth measurements on potted seedlings of trees, (4) ran the experiments for at least one month (i.e., between 35 and 50 days) and/or (5) used different tree populations of different latitudinal origins [14–16,20–23]. The treatments corresponded to day extensions with different R:FR ratios and main light periods of 9–12 h with similar light quantities (W m$^{-2}$) during the day and day extension/night treatment. Growth was measured at the end of the experiments as the distance between the soil and apical bud (shoot elongation) or the elongation of the needles, depending on the study. To quantitatively compare the results among the experiments, the measured growth parameters (i.e., either needle elongation or shoot elongation), were analyzed by considering only the effect size, i.e., growth relative to the average growth under pure R light day extensions and, if the experiment included more than one ecotype, the average growth of the most southern ecotype investigated under pure R light day extensions. For the analysis, the effect of the different light quality treatments and the population origin on the measured growth variables was analyzed through forward selection and backward elimination on a single dataset, where both variables were included in a two-way analysis of variance (ANOVA) with light quality and latitudinal origin as fixed factors. All analyses were performed using R 3.6 [24].

## 3. Results

### 3.1. Light Quality Changes Throughout the Day

Our field radiation measurements under different weather conditions and time of the day showed a reduction of the blue (B) light proportion and an increase of the green (G), red (R), and far red (FR) light proportion from sunrise and sunset to the middle of the day (Figure 1A–D). The analysis of multiple days with either clear or overcast conditions throughout the year revealed quality changes induced by the weather conditions that can be of similar magnitude as the diurnal effects of the solar elevation angle on the B and R fraction of the spectrum (Figure 2). In the middle of the day, the presence of clouds increased the B fraction, depending on the cloud cover density and height, with a simultaneous reduction of the R fraction. Weather conditions had no significant effect on values on the R:FR ratio in the middle of the day.

### 3.2. Effect of Weather on Light Quality Changes at Low Solar Elevations Angles

The effect of the weather conditions on the light quality was significantly stronger at lower solar elevation angles. At solar angels below 20°, overcast conditions led to a significantly lower proportion of B light and a higher proportion of R light, while the effect was much weaker for G light (Figure 2). At solar elevation angles below 1°, close to 37 % of the incoming PPFD consisted of B light, while G light and R light accounted for 31 and 30% of the PPFD, respectively, independently of the weather conditions. During a clear sky after sunrise and before sunset (sun angles between 5 and 8°), an average of 40, 33, and 28% of the light was coming from B, G, and R light, whereas under cloudy conditions at the same solar elevation angles, the values for these light qualities were 34, 32, and 34%, respectively

(Figure 2). A strong effect of low solar elevation angles was also found on the R:FR ratio. At clear sky conditions, the average R:FR ratio at 10° of solar elevation angle was 1.2, while it was close to 1.0 on cloudy days, and decreased strongly at solar angles <10° (Figure 2).

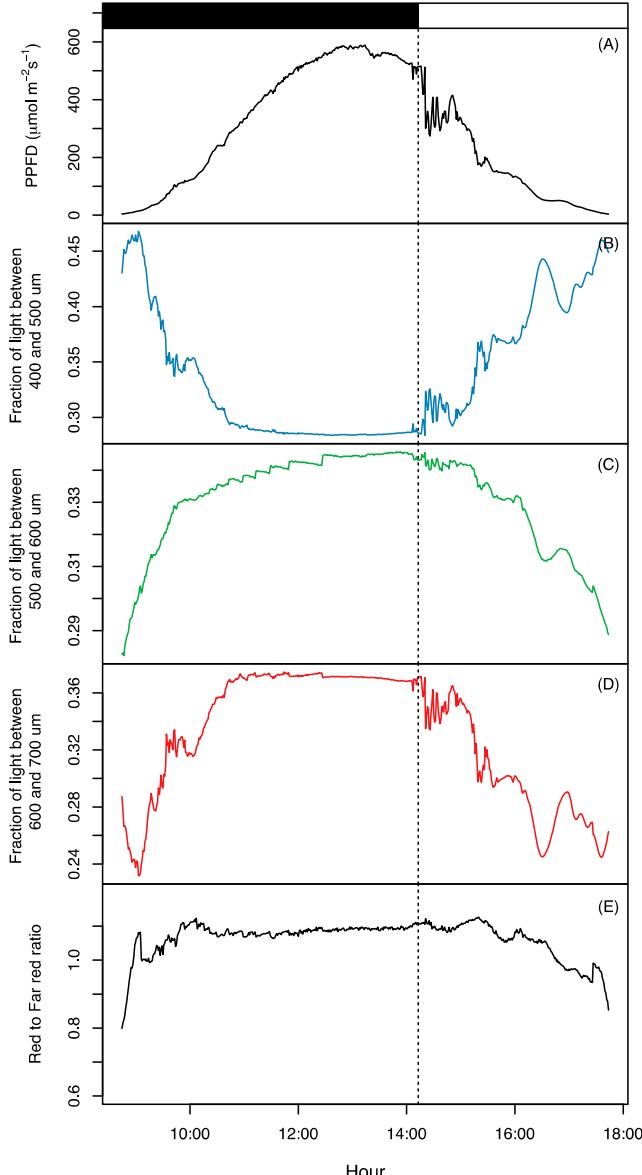

**Figure 1.** Changes in light quantity and quality as fraction of the photosynthetic photon flux density (PPFD) during a diurnal course. (**A**) Total PPFD; (**B**) blue light fraction (from 400 to 500 nm); (**C**) green light fraction (from 500 to 600 nm); (**D**) red light fraction (from 600 to 700 nm). (**E**) Red to far red (R:FR) ratio. The values are from a single, representative day with varying weather conditions with clear sky conditions until 14:15 (left hand side of the dotted vertical line) and partially overcast conditions during afternoon and evening (right hand side of the dotted vertical line). The data were recorded on 25 November 2018.

The inflection points calculated as the maximum value of the first derivate of each curve, for B, R light, and R:FR ratio under clear sky conditions was at a solar elevation angle of 13° for B and R and 14° for the R:FR ratio. Light quality quickly approached very stable values at solar elevation angles higher than 20°. As stated above, at solar elevation angles beyond 20°, only moderate effects of cloud cover on any wavelength fraction were found, with a small increase of the fraction of B light, on average, from 27% to 29% and a small reduction of the fraction of R from 38% to 36% at solar angles between 20

and 60° (Figure 2). The G light fraction, on the other hand, reached values close to 35% independently of the weather conditions, with its inflection point close to 10°. Finally, the R:FR ratio did not differ significantly between sunny and overcast days at solar elevations angles between 20° and 50° and stayed at a constant average value of 1.1, while at solar angles >50°, the R:FR ratio was even slightly higher on overcast days compared to days without cloud cover (Figure 2. More detailed values in supplementary Table S2).

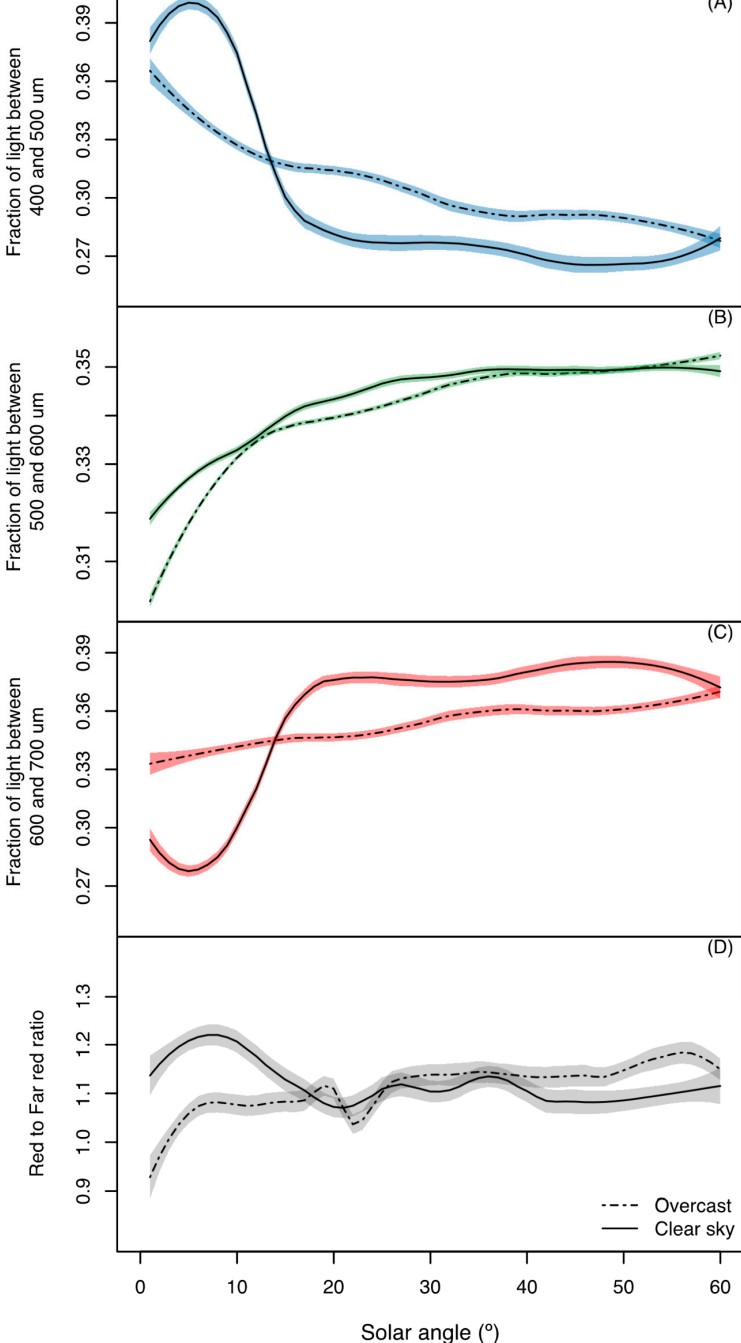

**Figure 2.** Changes in light quality as a fraction of the photosynthetic photon flux density (PPFD) depending of cloudiness (full line: Clear sky, dotted line: Overcast conditions) and the solar elevation angel. (**A**) Blue light fraction (from 400 to 500 nm), (**B**) green light fraction (from 500 to 600 nm), (**C**) red light fraction (from 600 to 700 nm). (**D**) Red to far red (R:FR) ratio. The lines represent the mean value of one day of each weather condition per month (*n* = 12; see methods for detail). Shaded areas correspond to the standard error of a locally estimated scatterplot smoothing (LOESS) fitted model.

### 3.3. Latitude Effects: Duration of Modified Light Quality and its Effect on Seedlings of Selected Tree Species

At higher latitudes, the period of daytime under modified sun light spectrum (i.e., solar angle below 20°) is significantly longer compared with lower latitudes, showing an exponential increase at higher latitudes. For example, at 30° N the maximum daily twilight period is reached as a single peak in mid-winter (Day 356) and does not exceed 5 h per day, while it shifts to the beginning of spring (Day 53) and the end of autumn (Day 292) at 60° N with a daily maximum duration of over 9 h (Figure 3).

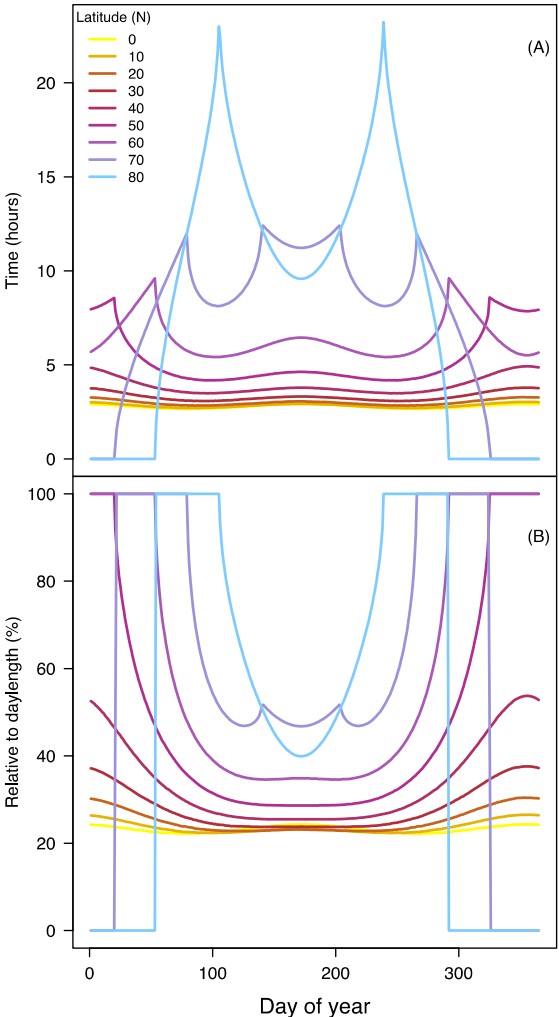

**Figure 3.** Estimated day length duration (**A**) and percentage relative to the total day length (**B**) of solar elevation angles between 0 and 20° for different latitudes.

For the seedlings of the selected tree species included in our analysis, both the light quality treatments and the latitudinal origin of the population significantly affected growth ($P_{value} < 2.2 \times 10^{-16}$ and 0.02, respectively). However, no interaction between these two factors was found ($P_{value} = 0.4$). The latitude effect is best described (fitted) as a quadratic effect (Figure 4A). Plants from higher latitudes had lower shoot elongation under the same light quality than southern ecotypes. In all studies higher R:FR ratios led to decreasing growth (Figure 4B). Additionally, the difference between the effects of the R and FR light treatments was more or less constant across trees from different latitudinal origin. For the light treatments, the best fit was a linear function after a logarithmic transformation, where plants treated with a larger fraction of FR light had larger elongation compared to trees treated with higher fractions of R light. Both factors were able to explain 82% of the variability (Figure 4. Available as 3D figure, Figure S1). A total of 54% of the variability was explained by the

light treatments and the origin of the ecotypes could explain 38% when the different variables were tested independently.

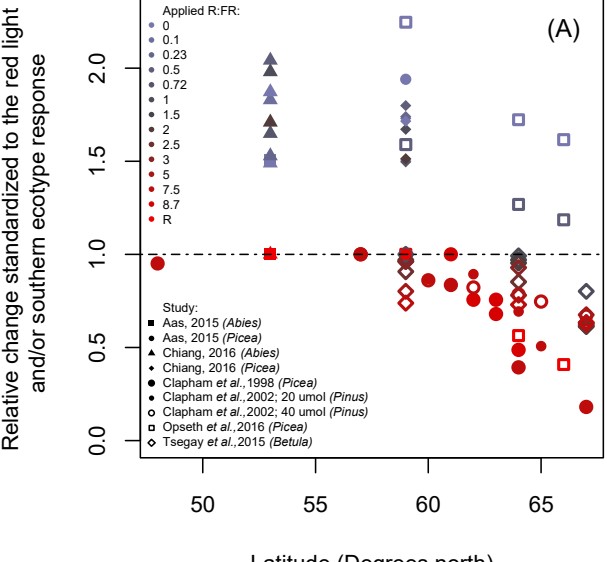

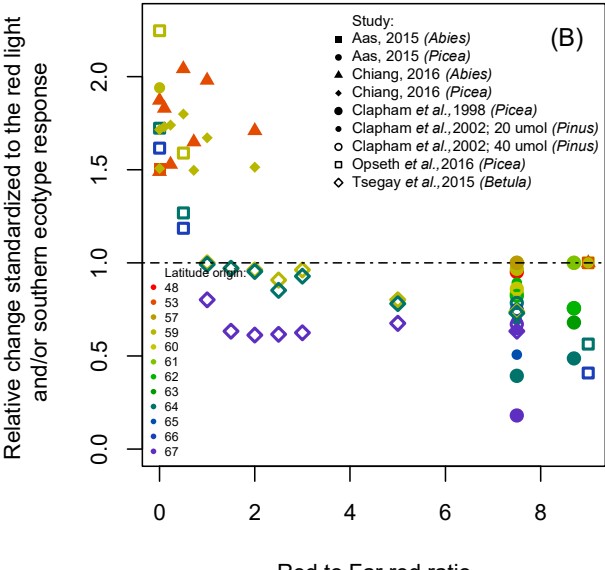

**Figure 4.** Effect of day light extension with different light qualities on seedling growth in selected temperate and boreal tree species: Relative changes of growth plotted (**A**) against the latitudinal origins under different red to far red ratios (R:FR ratios) and (**B**) against different R:FR ratios applied in trees from different latitudinal origins. The data were collected from work performed with seedlings of selected tree species (three evergreen conifers: *Picea abies*, *Pinus sylvestris*, *Abies lasiocarpa* [14–17,20–23] and one deciduous broadleaved species (*Betula pendula*) [19]), that were exposed to the different light quality treatments for 35–50 days. The additional legends give the first author, publication year, and tree genus. Data from Clapham et al. [23] were derived from two experiments with 20 and 40 μmol photons $m^{-2}$ $s^{-1}$ day extension light, respectively. The light treatments correspond to day extensions with different R:FR ratios and main day light periods of 9–12 h with similar light quantities in W $m^{-2}$. In the different experiments needle or plant height was measured as the growth response variable. Each study was standardized to the effect of red (R) light in the southern ecotype (when more than ecotype was included; see methods for details).

## 4. Discussion

Atmospheric constitution, e.g., the presence of clouds, can alter the composition of the light spectra. In our measurements, clouds increased the blue (B, 400–500 nm) light fraction by up to 10% in solar elevation angles above 20° through a reduction of direct light, which also led to a corresponding reduction of the red (R, 600–700 nm) light fraction in a similar magnitude. The green (G, 500–600 nm) light fraction was less affected by weather conditions, mainly due to a potentially 50% lower scattering compared to that of the B fraction [25]. R:FR ratios between weather conditions were not significantly different at high solar elevation angles but changed sharply at low solar angles.

It is well known that from solar elevation angles of −12° at the last two stages of twilight, i.e., the nautical and civil twilight (0° to −6° and −6° to −12°, respectively), the most substantial fraction of the spectra corresponds to the B light wavelength [1,12,26]. This is mainly due to a lack of direct radiation and a longer path length of the scattered sunlight through the atmosphere, which increases the probability of Rayleigh scattering of light by small atmospheric molecules and aerosols. With increasing solar elevation angles, there is an initial increase in the B light fraction together with a reduction of the fraction of R light (Figure 2). This initial increase in B light has been reported previously in a city environment at lower solar elevation angles than in our study [26]. This shift that was not present in a rural scenario, was explained by the presence of high-pressure sodium lamps as the city's main illumination source, were the timing for such a shift may accordingly depend on the city's illumination regime. The absence of this increase in rural scenarios and the low magnitude of this effect may indicate that this change should not play an important role as a biological signal in natural ecosystems. Once that the relative amount of direct light increases, a quick reduction of B light occurs, together with an increase of R light, mainly due to a shorter sunlight path length through the atmosphere that reduces the amount of B light refraction and therefore the B fraction in the sun spectra [2]. In contrast, G light tends to keep an asymptotic slower increase from lower to higher solar elevation angles with light quality reaching a steady state in solar angles higher than 10°.

The higher proportion of R light at twilight in cloud-covered conditions derives from the strong reflectance of R light from clouds into the lower atmosphere and the higher absorbance of B light by clouds. The intensity of this effect depends on the elevation of the clouds, its density, and its position on the horizon [27]. Many studies on radiation light quality with a more physical focus reported higher percentages of R light during sunrise and sunset than in the current study, mainly due to the direction of the used sensor and the aperture's angle. Zagury [27], for example, used a 25° aperture facing in the direction of the light source. This technical difference allows the sensor to detect exponentially more direct light and ignores the mostly diffuse light coming from other directions. The reduced amount of measured scattered light, which consists mostly of shorter wavelengths, therefore, increases the relative amount of R compared to FR light. Although the angle at which plants sense the light depends on the leaf angle, the measured values from the horizontal measurements and the inclusion of light coming from different angles as reported in our study here, are, on average, likely more similar to the light quality detected by plant leaves under natural conditions.

A very strong effect of the weather conditions at low solar elevation angles was found for the R:FR ratio, possibly partly explaining the larger differences between natural R:FR ratio measurements reported by previous authors (e.g., [1,28,29]). Although changes of the R:FR ratio in the presented magnitudes (Figure 2) have shown biological effects in short-term experiments with herbaceous plants [30], this effect has not been found in the few tree species investigated so far [15,16]. In contrast, many annual plants show high plasticity to lower fluctuation on the R:FR ratio conditions [31]. This may indicate that the previously investigated tree seedlings species may require several generations to adapt to the different R:FR ratios.

At higher latitudes, the period of daytime under modified sun light spectrum at twilight is exponentially longer compared with lower latitudes, especially during the spring and autumn. These prolonged periods of low solar elevation angle and the respective change of spectral light quality at higher latitudes might be used as pace-setting signals for plant biological processes in

perennial plants, like bud break, growth, or bud set. In woody plant species that have broad latitudinal distributions, such as the trees used in our literature review, ecotypes from southern latitudes have been shown to require less radiation to keep growing than conspecific ecotypes from the northern distribution edge. For example, Mølmann et al. [14] tested different radiation intensities and showed that the effect of FR and R light treatments also depended on light intensity and plant origin, e.g., 1.7 W m$^{-2}$ of FR light completely prevented bud set in more southern ecotypes (from 59 and 64° N latitude) of Norway spruce seedlings, while in a more northern ecotype (from 66.5° N latitude) only 43% bud set was reached. In all three ecotypes, lower light intensities, independent of the light quality, were not able to prevent bud set.

Several studies have suggested differential sensitivities of plant growth to light quantity and quality depending on their latitudinal origin [14–16,20–23] showing an interaction between the effect of light qualities and latitude [14,15]. The analysis of such data in the current study (see Figure 4A,B; Figure S1), also showed a distinct growth response to changed R:FR ratios depending on the latitudinal origin of the plants studied. The re-analysis of the combined data, clearly confirms the findings of the individual studies that an increased amount of FR light strongly promotes growth, compared to R light alone (Figure 4B). However, the previously observed interaction between the light treatments and the population origin was not found to be significant anymore ($P_{\text{value}} = 0.4$). Therefore, we propose that northern ecotypes tend to be as sensitive to light quality changes as southern ecotypes, but with higher light quantity requirements. The interaction between light quality and latitudinal origin on the growth of seedlings of temperate and boreal tree species proposed by previous authors, may actually be the result of two underlying, complementary responses: Firstly, an interaction between the light quantity and light quality (requiring higher amounts of FR light than R to reach similar results) and secondly, an interaction between the population origin and the light quantity (with northern ecotypes requiring more light than southern ecotypes). Remarkably, this is not contradictory to the results previously described by other authors, but a broader analysis may be needed to unequivocally reveal the relationship of the light quality and quantity requirements in tree seedlings from different latitudinal origins. Our analysis, together with our continuous light analysis suggests, that northern (or high latitude) ecotypes have adapted to longer photoperiods of modified light quality (mainly with respect to the R:FR ratio), which could have a more important role for biological processes like growth or bud set than in more southern (or low latitude) ecotypes of the same species. Although northern ecotypes may grow less under specific R:FR light conditions compared to southern ecotypes, at similar, non-saturating amounts of applied energy (in W m$^{-2}$), the amplitude of the effect of the light treatments between just R and FR light remain similar across the different latitudes (Figure 4A). This indicates that the accumulated amount of energy applied may play a more important role than the used light quality. Of course, effects of light quality and quantitiy on trees are occurring on top of other, fundamental environmental drivers, especially temperature were the effect of light quality have shown to be temperature dependent [13]. Thus, phenology and growth of trees species from boreal and temperate climates are regulated by temperature, light quantity, quality, and photoperiod, where the relative importance of each of these is likely dependent on the species origin latitude [32–35].

## 5. Conclusions

Here, we supply easily transferable continuous data of changes in light quality through the day and year in dependency of different weather conditions. These results are highly relevant from a plant biological perspective since the recorded wavelength areas are among the important determinants of plant growth and development. Such data is required to design LED systems simulating natural variation in light quality and quantity, which is becoming increasingly relevant today because an increasing number of plant growth facilities are using LED systems as the main source of radiation, and more natural light spectra are desirable.

Here, we also corroborate our hypothesis that the extended periods with modified light spectra at high latitudes correlates with the light requirements of seedlings of boreal and temperate tree species.

This suggest that in addition to other ecologically highly important factors such as temperature and photoperiod, changes in light quantity and quality play an important adaptive role on seedlings of woody plants at higher latitudes.

**Supplementary Materials:** The following material is available online at http://www.mdpi.com/1999-4907/10/8/610/s1, Figure S1: Effect of day extension with different red to far red ratios (R:FR ratios) on trees from different latitudinal origins, Table S1: Summary of the previous experiments used for the analysis, Table S2: Average light proportions of blue, green, and red and red to far red ratio under different solar elevation angles and two contrasting weather conditions.

**Author Contributions:** C.C., J.E.O., D.B. (Daniel Bånkestad), and G.H. designed the experiment; C.C. performed the experiment; C.C. and D.B. (David Basler) analyzed the data and prepared the figures; D.B. (David Basler) and J.E.O. validated the results; C.C. and G.H. wrote the initial draft of the paper. All authors provided inputs and suggestions and approved the manuscript for submission.

**Funding:** The present research was supported by PlantHUB - European Industrial Doctorate funded by the H2020 PROGRAMME Marie Curie Actions – People, Initial Training Networks (H2020-MSCA-ITN-2016). The program is managed by the Zurich-Basel Plant Science Center.

**Conflicts of Interest:** The authors declare no conflict of interest. The funders had no role in the design of the study; in the collection, analyses, or interpretation of data; in the writing of the manuscript, or in the decision to publish the results.

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
