# Peer review of "Latitude and Weather Influences on Sun Light Quality and the Relationship to Tree Growth"

_forests, doi:10.3390/f10080610_

Reviewer 1 Report

The manuscript “Latitude and Weather Influences on Sun Light Quality and The Relationship to Tree Growth” was improved as it was suggested by previous reviewers.

Reviewer 2 Report

Forests 2019 II , Revised version

Article:  Latitude and weather variation …

Chiang C., Olsen J.E., Basler D., Bankestad D., and Hoch G.

Final Comments for the authors

General remarks

This reviewer thinks that the article can now be published in a slightly revised version. On the one hand, it is a pity that the authors omitted the plant experiment with the LED lamp in Basel, but on the other hand, it is quite consequent and better to concentrate on a literature review and a documentation of own light quality and quantity measurements as was done in this revised version.

However, few improvements should still be made (see also Detailed comments). Authors should avoid generalizing too much: 1) For example, there is still only a selection of woody species, namely trees with needles from northern latitudes (only exception: Betula). Therefore, authors should avoid extending their results to ‘woody species’ in general as they do too often in the whole text. 2) There is certainly a hierarchy in the importance of ecological factors. Also in northern latitudes factors like temperature and light quantity are on top. Reading the text one gets the impression ‘R/FR’ are even more important or at least of the same importance. It would improve the quality of this manuscript if authors could avoid to stress the opinion that ‘R’ or ‘FR’ must be lifted up to the same position as temperature and PPFD (in total). It might be understandable that the authors like their aspect very much and therefore, tend to exaggerate. However, to do so is not really necessary.

Detailed comments

Abstract

line 19: The LED lamp does not play a role in the subsequent text. Why is it mentioned in the ‘Abstract’?

lines 32-33: This reviewer still insists very much on the deletion of the last part of this sentence from ‘…, while…’ on.

Material and Methods

line 135: LOESS: Give the full name!

line 147: Omit the word ‘extensive’ because it contradicts the statement that there is a lack of knowledge (as was emphasized before). It would also be better to call it a ‘literature review’ and not a ‘meta-analysis’. Normally, one needs a huge data base for an extensive meta-analysis and a lot of profound statistics.

lines 147-165: The reader needs some information about the tree species in the cited literature.  These authors probably concentrated on literature about trees with needles (conifers) from northern latitudes in Europe (exception: Betula). Therefore, it is some kind of a small selection from the huge world of woody species. It is urgently suggested to mention the origin of those species and give the names.

Discussion

line 292: Again ‘woody species’ in general (also in some other lines).

Conclusions

line 340-346: This paragraph is redundant because it is a mixture of ‘Abstract’ and ‘Material and Methods’ and contains none conclusion. Real conclusions start with line 346: ‘Finally, we…’.  ‘Conclusions’ should not have literature citations (line 346, line 352) but should concentrate on own findings in order to underline the novelty beyond already existing knowledge. The last sentence should be altered. The reader gets the wrong impression that ‘R/FR’ is of equal importance like temperature and PPFD (in total). Please pay attention to the hierarchy of ecological factors.

In total, ‘Conclusions’ have to be rewritten

Also the ‘Abstract’ has to be actualized.

References

This is only a random sample survey:

line 391: CO2, 2 as an index ↓.

line 429: ‘…  (Picea abies)…(Abies lasiocarpa)…’. Scientific names in italics!

line 432: ‘…  (Picea abies)… (Abies lasiocarpa)…’. Scientific names in italics!

Please correct also the references!

Reviewer 3 Report

Even though the authors use the term "meta-analysis" in their text, there is no application of this methodology. The authors use traditional methods of studies' review focusing on statistical significance testing (ANOVA), while in meta-analysis the research focuses on the direction and magnitude of the effects across studies. The term "effect size", i.e. the computation of a dependent variable that encodes the selected findings on a numerical scale, is mentioned only once in the manuscript, and it is never calculated in any way (standardized mean difference, odds - ratio, risk ratio etc.). There is no homogeneity analysis regarding the participating studies (Q - statistic), no funnel plot in order to examine the publication bias, and, finally, there is no forest plot for the overall assessment of the meta-analysis. The authors should apply a meta-analysis appropriately, or remove the term and adjust their text accordingly.

Author Response

Round  2

Reviewer 3 Report

The authors have satisfactorily addressed all the issues raised and made the necessary changes to the manuscript.

This manuscript is a resubmission of an earlier submission. The following is a list of the peer review reports and author responses from that submission.

Round  1

Reviewer 1 Report

A manuscript entitle “Latitude and weather variation on sun light quality and the relationship to woody plant phenology” was evaluated according to Forest Journal requirements. This manuscript provides an interesting information regarding to short- and long-term variation in light climate by measuring two major properties of light (quantity and quality). The reviewer found this manuscript very interesting.

Line 19, spell out the word LED and write the abbreviation between parenthesis.

Please write together the red to far red components F:FR ratio, space is not need between both factors, also a consistence across the entire manuscript for (R:FR) is requires.

Line 100, use the correct symbol to denote “degree” or declare what this symbol mean. Please use the correct one and fix everything across the entire manuscript.

Figure 1: find a way not to overlap the X-axes values between each panels E) and D): e.g numbers can go from 0.6 to 1.8 panel E) and from 0.24 to 0.32, choose the best way no to created confusion to readers.

Figure 2: it has the same problem as figure 1, but here is more obvious how panels are overlapping between each other.

Lines 285-286, If there was statistical analysis, please provide the p-value for this statemen.

Lines 312-315, Conclusion should be clear and convincing rather than a speculation. Please fix the entire sentence.

Reviewer 2 Report

Forests 2019 I.

Article:  Latitude and weather variation …

Chiang C., Olsen J.E., Basler D., Bankestad D., and Hoch G.

Comments for the authors

General remarks

Studies about light quality effects on the performance of woody plants are rather rare according to the knowledge of this reviewer because until now it is partly outside of the main focus in eco-physiological research. Therefore, the most important value of this paper is the very detailed documentation of this still somehow neglected factor in nature. For this reason, this information should be available to the scientific community reading ‘Forests’. Nevertheless, there is one strong weakness that should be avoided by the authors: The text is mainly concentrated on the values of measurements of light quality and quantity, and readers do not learn enough about the response of plant species, of the performance of ecotypes in reaction upon those factors, and also they do not read enough about the planting design in the Botanical Garden of Basel (Switzerland). For instance, what means Aas-Abies, etc.? Did the plants shade each other? Why does Betula appear in Figure 4 and its legend says that this species was excluded? Why is the time of bud-break not documented? Why is height growth not presented in a more detailed manner? Thus, overall there is not much phenology in this paper although this is announced in the title. Further questions see under ‘Detailed comments’.

The authors could also think – at least in ‘Discussion’ – a little about the effect that the light transmitted through the leaves in the canopy above and reaching the seedlings underneath on the ground is not only quantitatively but presumably also qualitatively changed.  The ecological needs of the species are not considered sufficiently enough.

Detailed comments

Title

line 3: Perhaps the word ‘phenology’ should be omitted because data were only rather  indirectly  or theoretically linked  to phenology.

Abstract

20:      Why ‘but’. Sense is not quite clear.

20-21: Redundant! Please avoid twice ‘easy to apply’.

33-34: Speculations about the influence of other factors should be avoided because they were not on focus in this study. (Wavelength of solar radiation was only measured up to 1000 nm.)

Keywords

35:   Light quality ,…

Introduction

41-43: Also this reviewer thinks that such a study was not done before intensively enough. The reviewer could only find one very old booklet with a somehow similar intension: Overdieck, D., 1978: Wirkungen der sichtbaren Strahlung auf CO2-Gaswechsel  und Transpiration höherer Landpfanzen. Habilitationsschrift, TU-Beriln, Germany. Unfortunately, it is in German but seems to have an English ‘Extended Summary’.

71:       What about the influence of the canopy above in the forests?

79-80: Here, the interaction of temperature and light quality in lower wavelength is not obvious to the reader.

87-89: This last sentence should be omitted because it is rather speculative.

At the end of ‘Introduction’ it would be nice to find clearly formulated working hypotheses in form of questions which should be answered later-on.

Material and Methods

136-145: It is wished (even essential) to know at least something about the species used in this study. How were they selected? How were they planted? How were bud break and other phenological features determined? How was plant growth measured? Why was Betula pendula excluded and perhaps replaced by another Betula in Figure 4? Was the experimental situation really comparable with that in the forests, where seedlings are growing underneath of the canopy of the oldies? This reader urgently needs more details here.

Results

147-156: Light quality changes are well documented.

157-177: Again the question could be started: What about light quality and quantity after transmission through leaves and in flushing light-flecks on the ground?

Fig. 1: Why did the authors chose an example from November (!) after the end of the growth season? Does this make sense?

137:         What means ‘loess’?

195-204: Without enough knowledge about the ecotypes this paragraph is not understandable.

Fig. 4 and

209-216: As already mentioned several times, the ecotypes appearing in the graph should be explained in detail. The listed species are rather different in their environmental demands. For instance, juvenile Abies alba (Was it this one?) is shade-tolerant whereas young Pinus sylvestris (Was it this one?) prefers more sun-light. Why does Tsegay (?)-Betula have symbols in the graph on the one hand,  and it is, on the other hand,  somehow excluded by the legend underneath? All species and ecotypes should be clearly defined at least here.

Discussion

240 pp.: The most important message of this manuscript is well discussed (latitude → light quality). Link to plant performance is partly missing.

252-253: …therefore,… In addition, the sense of this sentence is not quite clear.

264.      and not necessarily transfer this information to the next generation. … Please omit!

Hypotheses and questions presented in ‘Introduction’ should be clearly answered at the end of ‘Discussion’.

Conclusions

314-315:  It is probably better to keep ‘Conclusions’ free from speculations and concentrate on what is clarified by the study itself.

Author Contribution

320:           This reader would prefer to see the full last names of the contributors here.

Overall summary of the reviewer

Bud set/burst and growth data are only shortly mentioned but not explicitly enough presented. One expects to get those by reading the title. Plant species and ecotypes are not described sufficiently enough. The idea to use LED light in the experiment and to simulate the light climate of latitudes is original and is worthwhile to be published. (In this context, a very exact comparison of sun-light and LED-light (quality?) would improve the manuscript essentially.) However, this is not yet linked intensively enough to plant performance. Therefore, major improvements have to be suggested.